# Liquid–Liquid Phase Separation in Cancer Signaling, Metabolism and Anticancer Therapy

**DOI:** 10.3390/cancers14071830

**Published:** 2022-04-05

**Authors:** Sebastian Igelmann, Frédéric Lessard, Gerardo Ferbeyre

**Affiliations:** 1Department of Biochemistry and Molecular Medicine, Université de Montréal, Montréal, QC H3C 3J7, Canada; sebastian.igelmann@umontreal.ca; 2Montreal Cancer Institute, CR-CHUM, Université de Montréal, Montréal, QC H2X 0A9, Canada; 3Laboratory of Growth and Development, St-Patrick Research Group in Basic Oncology, Cancer Division of the Quebec University Hospital Research Centre, Québec, QC G1R 2J6, Canada; frederic.lessard@umontreal.ca

**Keywords:** biomolecular condensates, liquid-liquid phase separation, metabolon

## Abstract

**Simple Summary:**

Emerging evidence shows that the organization of the cells into membraneless sub-compartments confers unique properties to cancer cells. The biochemical and biophysical mechanisms underpinning the formation of these compartments, also called biological condensates, enlighten how some mutations and gene expression changes contribute to cancer formation and progression.

**Abstract:**

The cancer state is thought to be maintained by genetic and epigenetic changes that drive a cancer-promoting gene expression program. However, recent results show that cellular states can be also stably maintained by the reorganization of cell structure leading to the formation of biological condensates via the process of liquid–liquid phase separation. Here, we review the data showing cancer-specific biological condensates initiated by mutant oncoproteins, RNA-binding proteins, or lincRNAs that regulate oncogenic gene expression programs and cancer metabolism. Effective anticancer drugs may specifically partition into oncogenic biological condensates (OBC).

## 1. Introduction

The regulation of cellular homeostasis through metabolism and cell signaling requires both short-term adaptations to stresses and long-term stable changes in cellular organization. Examples of this spatiotemporal control are heat shock condensates, stress granules, P bodies and RNA condensates. In the assembly and disassembly of these biomolecular condensates it is possible to rapidly alter the response of a cell towards a stress and achieve a fine tuning of complex biochemical reactions. In addition to rapid responses, condensates form membraneless organelles in which several biological processes can be optimized and stably inherited [1,2]. During development, asymmetric partitioning of protein and RNA condensates determine the cells that become the germ line and may also control other developmental decisions [3]. In this way, biological condensates provide a novel paradigm to understand non-genetic mechanisms controlling cell fate.

Biomolecular condensates form via liquid–liquid phase separation (LLPS), a process favored by the energy minimization within coexisting liquid phases that results in the formation of distinct liquid droplets. As a result, the biophysical properties of the droplets are different from its surrounding dilute phases and exhibit a higher viscosity [4,5]. The process of LLPS is favored by multivalent interactions between specific protein domains, intrinsically disordered regions in proteins (IDR) and nucleic acids [4]. Fluorescence recovery after photobleaching (FRAP) is a standard technique used to demonstrate the liquid-like property of biomolecular condensates. In this technique, the biomolecule under study is fused with a fluorescence label and a laser is used to bleach the fluorescence signals visualized as condensates using a fluorescent microscope [5]. LLPS is commonly reversible, and the resulting droplets can undergo fusion and fission. However, some droplets can transit to solid aggregates, often causing cell dysfunction [6,7]. Solid aggregates do not recover their fluorescence in FRAP experiments quickly [5]. Additional proof for a phase separation mechanism should be provided by quantitative measurements to demonstrate a concentration dependence for the formation of biomolecular condensates. Many studies provide this proof in vitro but this might not be relevant for the actual protein concentrations in cells [8]. The protein concentration in phase-separated structures depends on the ability of each protein to engage in intermolecular interactions. For example, the RNA-binding protein DDX4 forms condensates at a concentration close to 5 mM but a mutant of this protein that cannot form interactions does not separate under the same conditions [9]. A direct measurement of the concentration in the condensates using the ^1^H-^15^N heteronuclear quantum coherence spectrum showed a much higher value of 27.7 mM for the RNA-binding protein FUS [10]. A similar concentration (around 14 mM) was obtained using Raman spectroscopy of condensates formed by ataxin-3, a protein that forms aggregates in the neurodegenerative Machado–Joseph disease [11]. The total protein concentration in the cells has been estimated to be around 2.5 mM [12]. Therefore, the degree of crowding in biomolecular condensates is not too different from the rest of the cells but the effective increase in concentration for each phase-separated protein is considerable.

Biological condensates affect cell functions via several mechanisms. First, condensates increase the rate of biochemical reactions due to the high local concentration of enzymes, substrates and interaction partners [13,14]. Signaling complexes that assemble via weak interactions are particularly favored in biomolecular condensates as demonstrated by the T cell and B cell receptor complexes [15,16]. Second, condensates can enhance reactions by excluding inhibitors [16]. Third, they can buffer biomolecule concentrations for which levels change due to noise in the biological process [17]. Fourth, some condensates provide specific pathways for molecular transport and protein sorting [18,19]. For example, nuclear pore proteins form hydrogels that block the transport in and out of the nucleus for many macromolecules [19]. Finally, biomolecular condensates limit certain reactions by sequestering key components [20]. Many different physiological processes are thought to depend on LLPS, including the organization of rRNA synthesis and processing in the nucleolus [21], RNA metabolism in stress granules [22,23,24,25] and paraspeckles [26], the organization of the chromatin [27], transcription [28,29] and DNA repair in DNA damage foci [30,31].

Aberrant phase separation and liquid to solid transitions have been well demonstrated in several neurodegenerative diseases [32]. However, emerging evidence also indicates that altered phase separation underpins cancer cells’ phenotypes. Many mutations affect proteins and RNAs in cancer cells but only a few directly change the activity of these molecules. An emerging concept is that cancer mutations could change the ability of macromolecules to form biomolecular condensates, indirectly affecting their activity [33]. In addition, biomolecular condensates can provide a robust mechanism of spatial memory within cancer cells [3], which can potentially mediate non-genetic mechanisms explaining tumor heterogeneity and drug resistance.

### 1.1. Phase Separation and Mutant Oncoproteins

Cancer-associated mutations can promote the formation of new biomolecular condensates. Phase separation in mutant oncoproteins can be explained by the ability of the mutant proteins to engage in multivalent interactions. Several examples of this phenomenon have already been identified, controlling both cell signaling and transcription (Figure 1).

Cancer-associated SHP-2 (SH2 Domain-Containing Protein Tyrosine Phosphatase 2) mutants recruit the wild-type SHP2 protein into phase-separated condensates that promote the activation of oncogenic RAS-MAP kinase signaling (Figure 1A) [34]. SHP2 does not contain an IDR or repetitive domains that can engage multivalent interactions. Rather, droplet formation was dependent on multiple negatively and positively charged residues mediating electrostatic interactions and was highly sensitive to high salt concentrations [34]. FRAP analysis showed that these droplets recovered within minutes after photobleaching, which was consistent with their liquid nature. Another oncoprotein that undergoes LLPS is the fusion protein EML4-ALK (Echinoderm Microtubule-Associated Protein-Like 4-Anaplastic Lymphoma Kinase), which is present in lung cancer. This fusion protein localizes in punctate cytoplasmic structures that act as a platform to activate the RTK/RAS/MAPK pathway (Figure 1A). The biophysical properties of EML4-ALK bodies are not consistent with LLPS structures since they are resistant to the aliphatic alcohol 1,6-hexanediol and exhibited porous curvilinear shapes in super-resolution images revealed by structured illumination microscopy (SIM). FRAP showed a slow recovery for most of the EML4-ALK bodies, consistent with a highly viscous state. The EML4 portion of the EML4-ALK fusion protein contains an N-terminal trimerization domain that is required for granule formation and RAS/MAPK activation, demonstrating that the generation of membraneless structures is critical for the activation of oncogenic MAPK signaling by this mutant oncoprotein (Figure 2A) [35]. It is worth considering why these oncoproteins act by phase separation rather than simply providing a scaffold or MAPK activation as reported for Ste5 and KSR1 [36].

Signaling scaffolds facilitate protein–protein interactions and regulate crosstalk with other signaling pathways [36]. Phase separation must be conferring additional advantages over scaffolding for these signaling modules to play a role in the transformation. In this sense, it is worth considering that aberrantly high activation of the ERK pathway induces senescence rather than oncogenic transformation [37,38]. Phase-separated signaling complexes may allow for a moderate level of signaling output, stimulating proliferation while avoiding the aberrantly high activation levels triggering senescence (Figure 2B). One possibility is that phase separation may exclude phosphatases from signaling complexes, as reported for T cell receptor signaling [16]. The competition of phosphatases with kinases in signaling complexes confers ultra-sensitivity in signaling pathways [39]. Ultra-sensitivity amplifies the signal and can mediate changes in cell fate by allowing a higher level of signaling output [40]. Phase-separated MAPK signaling complexes organized by mutant SHP2 or EML4-ALK may exclude phosphatases and avoid ultra-sensitivity in ERK signaling, preventing the levels of signal amplification required for senescence.

Another cancer-associated mutation that alters signaling pathways occurs in the tumor suppressor NF2 (Neurofibromin 2). The mechanism by which NF2 mutations drive cancer is unknown but recent results show that mutation in the FERM domain of NF2 lead to a gain-of-function ability to form biomolecular condensates that sequester and inactivate IRF3 and TBK1, disrupting the cGAS-Sting pathway (Figure 1B) [41]. The NF2 condensates exhibited a quick recovery after photobleaching, suggesting that the sequestration model for the cGAS-Sting pathway is not absolute. In fact, this pathway also acts as a double-edged sword in cancer. In premalignant cells, cGAS-STING mediates oncogene- induced senescence, opposing transformation [42,43]. However, many tumors express high levels of genes in the cGAS/STING pathway [44,45]. Again, phase separation may adjust the output of cGAS/STING signaling, avoiding senescence while still conferring an advantage to the tumor cells.

Cancer-associated mutations can also drive the formation of phase-separated transcription complexes that drive the expression of oncogenic transcription programs (Figure 1C). Mutation in the YEATS domain of ENL (Eleven-Nineteen Leukemia) in Wilm’s tumors enhances the ability of this protein to seed biomolecular condensates at the loci with acetylated chromatin, recruiting transcription complexes to activate gene expression [46]. ENL-mutant condensates recovered quickly after photobleaching, revealing their dynamic nature. Similarly, the fusion proteins between the N-terminal phenylalanine-glycine rich (FG)-repeat domain of the nuclear pore protein NUP98 with the chromatin binding domains of HOXA9, KDM5A or NSD1 also increase the transcription of oncogenic programs by promoting the formation of biomolecular condensates and super-enhancers that recruit transcriptional coactivators [47]. Finally, the EWS-FLI1 fusion protein present in Ewing Sarcoma recruits the chromatin-remodeling complex BAF to tumor-specific enhancers activating gene expression via LLPS [48]. EWSR1, one of the partners in the EWS-FLI1 fusion proteins, requires RNA to interact with the BAF complex. However, the EWS-FLI1 fusion protein forms transcriptional activating biomolecular condensates that do not require RNA [48]. One important question is why phase separation is needed for transformation mediated by these mutant oncoproteins. One can speculate that cancer mutations in biomolecular condensates change the biophysical properties of condensates, making them more stable. As a consequence, cells expressing these oncoproteins cannot reverse the activation of gene expression programs that maintain the cancerous stage, which are similar to programs that characterize embryonic or stem cells [49,50,51,52]. This idea suggests that drugs capable of dissolving condensates of mutant oncoproteins could inhibit tumorigenesis.

### 1.2. Phase Separation and Tumor Suppressors

Cancer mutation can also disrupt the normal functioning of biomolecular condensates that prevent tumorigenesis. The functions of p53, the most frequently altered tumor suppressor in human cancer, are inactivated in phase-separated amyloid structures for which formation is facilitated by mutations in the DNA-binding domain of p53 [53]. The tumor suppressor E3 ligase SPOP forms phase-separated bodies in the nucleus with its substrates proteins such as DAXX. Cancer-associated mutations of SPOP disrupted the SPOP/DAXX nuclear bodies, driving DAXX accumulation and cancer progression [54]. cAMP compartmentation, a process that depends on the phase separation of RIα and PKA, is disrupted by mutations of RIα present in liver fibrolamellar carcinoma (FLC). RIα is the regulatory subunit of PKA, a protein kinase that transduce cAMP signaling. The consequence of this RIα mutation is an aberrant cAMP signaling that increases cell proliferation and induces malignant properties in non-cancerous liver cells (Figure 3A) [55].

The promyelocytic leukemia protein PML acts as tumor suppressor, regulating apoptosis and cellular senescence [56]. PML forms phase-separated nuclear bodies, for which composition is regulated by multivalent interactions mediated by SUMOylation [57,58]. PML bodies organize the repression of the E2F target genes required for cell proliferation by organizing repression compartments with the retinoblastoma tumor suppressor [59,60]. In acute promyelocytic leukemia, the fusion protein PML-RARA disrupts the PML body’s function, promoting transformation [61]. Another tumor suppressor that requires phase separation for its functions is the histone demethylase KDM6A, also known as UTX. The most frequent cancer mutations in UTX affect its IDR, abolishing the ability of the protein to form biomolecular condensates and regulate high-order chromatin interactions [62]. Intriguingly, the tumor suppressor functions of UTX do not depend on its demethylase activity but on phase separation, but we know very little about how UTX condensates actually block tumorigenesis. The enigmatic functions of both PML and UTX suggest that their tumor suppressor activities could be linked to the disruption of oncogenic biomolecular condensates (Figure 3A).

### 1.3. RNA-Binding Proteins and Non-Coding RNAs That Affect LLPS in Cancer

Biological condensates driving tumorigenesis can also be generated by an increased expression in proteins or RNA capable of undergoing LLPS. LLPS often depends on multivalent interactions that involve RNA molecules and/or RNA-binding proteins [24,26,63,64]. The RNA-binding protein AKAP8 (A-Kinase Anchoring Protein 8), also known as AKAP95, is overexpressed in ovarian and colorectal cancers and is required to inhibit oncogene-induced senescence. The oncogenic activity of AKAP8 involves LLPS and correlates with its ability to regulate RNA splicing [65]. For example, AKAP8 expression and LLPS leads to intron retention during the splicing of the cyclin A2 transcript. The resulting cyclin A2 isoform is resistant to degradation, driving oncogenic transformation [65]. Intriguingly, RNA binding reduces the mobility of AKAP8 in condensates [65]. Of note, AKAP8 prevents metastasis in breast cancer, indicating that the pro-tumorigenic effect is context-dependent (Figure 3B) [66].

The RNA-binding protein YBX1 (Y-Box Binding Protein 1) has emerged as an important driver in multiple cancers. YBX1 forms biomolecular condensates that recruit miR-223, sorting this miRNA to exosomes [67]. YBX1 also regulates transcription, splicing and translation, and is required for viability in cells that persist after targeted therapy against Jak2 [68,69]. Persistence is a non-genetic mechanism of the adaptation to stress that could be mediated by the formation of stable biomolecular condensates [8]. In sarcomas, YBX1 increases the translation of the RNA-binding protein G3BP1, which, via LLPS, initiates stress granule formation that enables cancer cells to sustain stress (Figure 3B) [70].

SNHG9 (Small Nucleolar RNA Host Gene 9) is a lipid-binding lncRNA highly expressed in breast cancers. Its oncogenic activity was demonstrated to be dependent on the sequestration of the tumor suppressor LATS1 via LLPS. LATS1 is a member of the Hippo pathway deregulated in multiple cancers [71]. The RNA-binding protein, YTHDC1, drives tumorigenesis in acute myeloid leukemia by protecting oncogenic mRNAs from degradation. This protein is a reader of the epigenetic RNA modification N^6^-methyladenosine. The modification allows YTHDC1 to form biomolecular condensates called YACs (YTHDC1-m^6^A condensates), which are increased in leukemia cells (Figure 3B) [72].

### 1.4. Phase Separation and Cancer Metabolism

Enzymes involved in specific metabolic pathways are often organized into multi-enzymatic complexes or metabolons. This strategy can increase the activity of enzymes with poor specificity or low substrate affinity, or prevent the accumulation of toxic metabolites for which their only role is to act as intermediates in a metabolic pathway [73]. The compartmentalization of metabolic pathways also allows substrate channeling where the metabolic intermediates are not in equilibrium with the bulk solution and the reactions do not depend on the free diffusion of these metabolites from long distances [74].

The enzyme ribulose bis-phosphate carboxylase oxygenase (RuBisCO) is found in 100–200 nm polyhedral bodies known as carboxysomes in bacteria [75] or in non-membrane organelles called pyrenoids in green alga [13],. In this way, carbon fixation is optimized perhaps by shielding RuBisCO from the actions of oxygen [75]. Both in bacteria and eukaryotes, RuBisCO adopts a liquid-like state, the assembly of which is mediated by multivalent interactions mediated by accessory proteins [13,76,77]. In yeast, a large number of metabolic enzymes form punctate cytoplasmic foci upon nutrient depletion [78]. Therefore, biological condensates may play key roles in stressed cells that require specific metabolic pathways to survive.

Consistent with the idea that metabolic stress leads to increased formation in biomolecular condensates, mammalian cells under purine-depleted conditions form purinosomes, where all enzymes for de novo purine biosynthesis cluster in puncta [79]. Intriguingly, hypoxia triggers the formation of purinosomes in a process that requires the activity of the transcription factor HIF1α [80]. Hypoxia also triggers the formation of discrete bodies of glycolytic enzymes (G bodies) [81]. The formation of G bodies requires RNA and multivalent interactions, suggesting they form via LLPS [82]. Importantly, G bodies also form in human liver cancer cells in response to hypoxia [81]. Additionally, human PFK (Phosphofructokinase), which catalyzes the rate-limiting reaction in glycolysis, localizes in discrete cytosolic clusters in liver cancer cells together with other enzymes of glucose metabolism. Of note, these foci were not present in non-cancerous cells, suggesting that they play a role in cancer metabolism [83]. Hypoxia also triggers the formation of puncta containing pyruvate carboxylase, malate dehydrogenase-1 and malic enzyme-1. These enzymes form a complex called HTC (hydride transfer complex) that catalyzes the transfer of hydride ions from NADH and NADP, regenerating NAD+, and forming NADPH (Figure 4). HTC assists the metabolism of cells with mitochondrial dysfunction and contributes to malignant transformation by inhibiting the process of oncogene-induced senescence. HTC puncta are dissolved by 1,6 hexanediol, suggesting that the complex is formed via LLPS [84]. Finally, amino acid starvation triggers LLPS of the proteasome, a process that decreases cell survival during p53-mediated apoptosis [85].

Another biomolecular condensation that facilitates tumorigenesis is initiated by the LLPS of glycogen in liver cancer. Glycogen accumulates in liver cancer cells due to the downregulation of the last enzyme in glycogenolysis, glucose 6-phosphatase. The glycogen condensates sequester and inhibit the tumor suppressors, Mst1/2, in the Hippo pathway and as a consequence, they stimulate YAP-dependent tumorigenesis. The carbohydrate-binding protein Laforin was found to be responsible for transporting Mst1/2 to the glycogen condensates [86]. Finally, cancer cells are motile and require high concentrations of ATP and GTP at lamellipodial and filopodial projections. Enzymes in GTP and ATP biosynthesis are localized at these compartments in cancer cells to meet these metabolic requirements and it is likely that LLPS plays an important role in this process [87].

### 1.5. Phase Separation in Cancer Therapy

Since phase separation is often mediated by IDRs, compounds that bind these domains may help to modulate the process. However, few compounds have been identified that bind IDRs and their ability to disrupt biomolecular condensates is poorly explored [88]. Drugs may localize or be excluded from biomolecular condensates, affecting their ability to reach their targets. For example, cisplatin, mitoxantrone, the CDK7 inhibitor THZ1 and tamoxifen were found to selectively partition into condensates formed by MED1, a component of transcriptional condensates (Figure 5A). Moreover, mitoxantrone concentrated in condensates formed by the nucleolar proteins FIB1 and NPM1 (Figure 5B). On the other hand, the BRD4 inhibitor concentrated in condensates of BRD4, ED1 and NPM1 (Figure 5C).

The ability of platinum drugs to concentrate in condensates of MED1, a component of super-enhancers, may explain why platinum drugs are efficient in many cancers [89]. Intriguingly, MED1 overexpression confers resistance to tamoxifen in breast cancer and this was explained by an increase in the size of transcriptional condensates and an inability of tamoxifen to prevent the partitioning of ERα in these condensates [89]. The importance of intracellular localization for anticancer drug action is well illustrated in studies using multiplexed ion beam imaging (MIBI) and Nano SIMS (nanoscale secondary ion mass spectrometry) instruments in cells that develop resistance to cisplatin. In sensitive cells, cisplatin is enriched in nuclear speckles and excluded from closed chromatin. However, in resistant cells the drug is totally excluded from the nucleus (Figure 5D) [90]. Another potentially useful technique to study the effects of drugs at the single-cell level is Raman spectroscopy. The Raman spectral signature of cells changes after treatment with drugs such as tamoxifen, and some Raman peaks correlate with the tamoxifen abundance in the same cells as measured using mass spectrometry [91]. It will be interesting to determine whether changes in the Raman peaks of drugs in cells indicate their accumulation in phase-separated structures. Nevertheless, the new data show that the ability of drugs to reach targets requires not only the crossing of biological membranes, but also the partition into membraneless condensates. It is important to consider these capabilities during drug development.

## 2. Conclusions

The formation and stability of oncogenic biomolecular condensates (OBC) add a new dimension to cancer biology beyond the classic genetic models commonly used to explain the disease. OBC may define novel exploitable vulnerabilities in tumors. Further work using imaging mass spectrometry to obtain nanoscale-resolution images of drug localization will be important to determine to what extent intracellular drug partitioning is important for their efficacy.

## Figures and Tables

**Figure 1 cancers-14-01830-f001:**
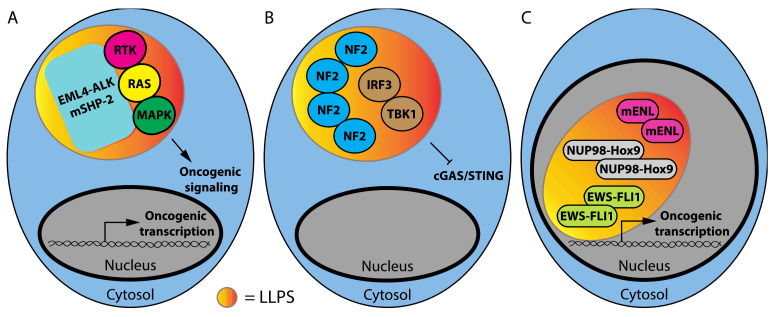
Biological condensates of mutant oncoproteins. (**A**) SHP-2 and EML4-ALK roles in LLPS formation and RAS-MAP kinase signaling. (**B**) NF2 roles in LLPS formation, sequestration of IRF3/TBK1 and disruption of the cGAS-Sting pathway. (**C**) LLPS of transcription complexes as driver of transcription programs.

**Figure 2 cancers-14-01830-f002:**
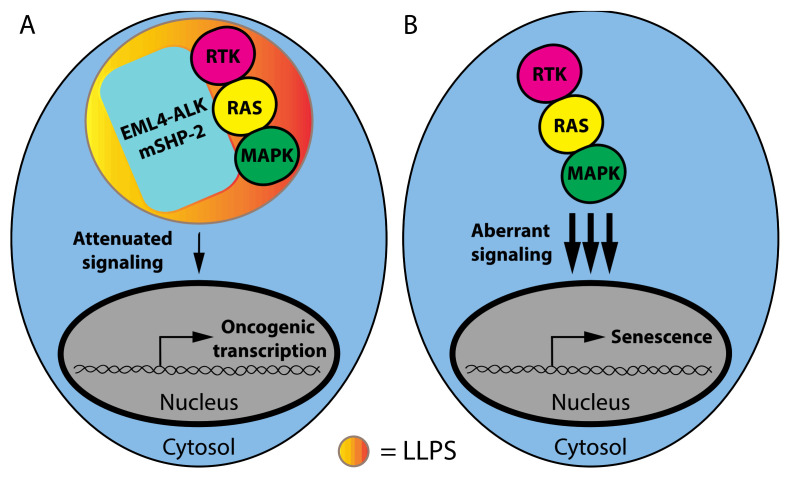
Phase separation attenuates oncogenic signaling, avoiding cellular senescence. (**A**) LLPS implication in attenuation of RAS-MAP kinase signaling. (**B**) LLPS absence leading to aberrant RAS-MAP kinase signaling and the senescence program.

**Figure 3 cancers-14-01830-f003:**
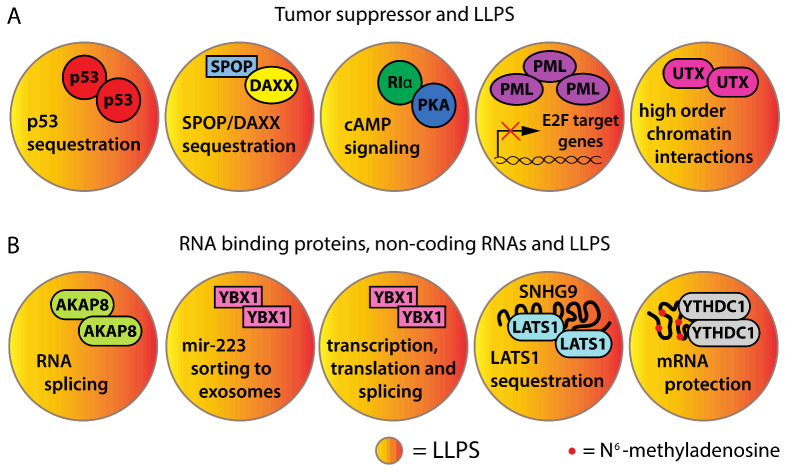
Tumor suppressor and RNA implication in LLPS. (**A**) Roles of tumor suppressors and/or their associated mutations in LLPS formation and downstream signaling in cancer. (**B**) Roles of RNA-binding proteins and non-coding RNAs in LLPS formation and downstream signaling in cancer.

**Figure 4 cancers-14-01830-f004:**
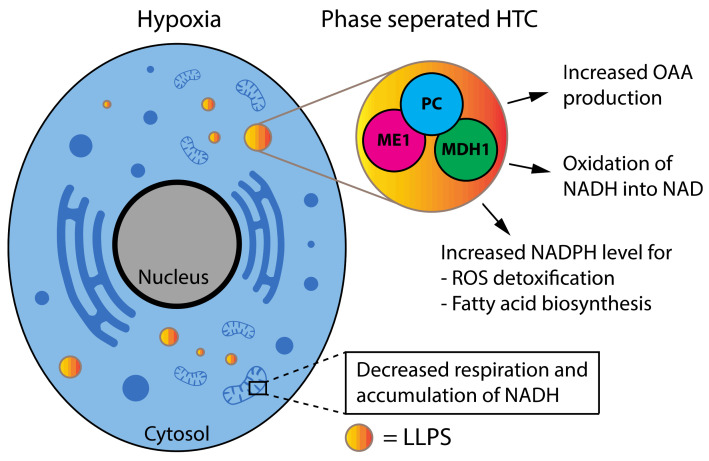
Hydride transfer complex (HTC), a phase-separated oncometabolon. Under hypoxia ME1, PC and MDH1 form a complex in LLPS leading to OAA production, oxidation of NADH into NAD and increased NADPH level.

**Figure 5 cancers-14-01830-f005:**
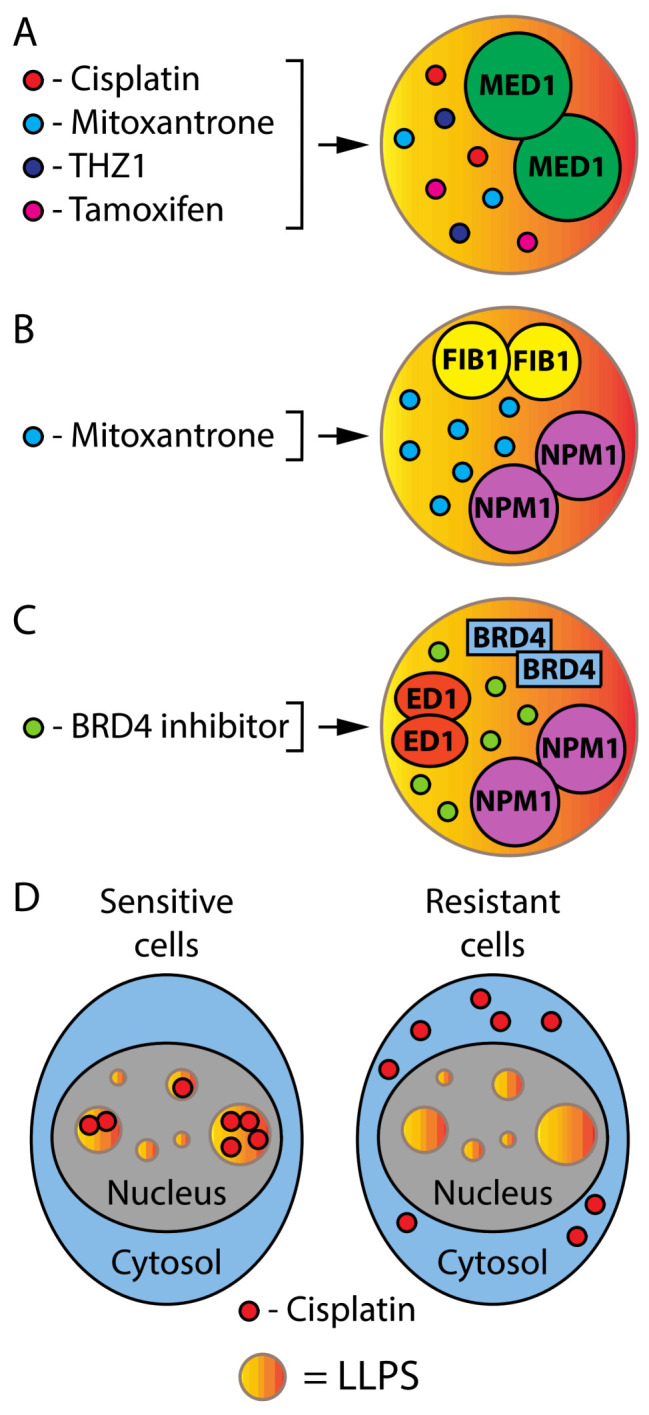
LLPS and cancer treatments. (**A**) Drugs found in LLPS of MED1. (**B**) Mitoxantrone can be found in LLPS of FIB1 and/or NPM1. (**C**) The BRD4 inhibitor can be found in LLPS of BRD4, ED1 and/or NPM1. (**D**) Cisplatin localizes in nuclear LLPS of treatment-sensitive cells but in the cytoplasm of treatment-resistant cells.

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
