# Peer review of "Liquid–Liquid Phase Separation in Cancer Signaling, Metabolism and Anticancer Therapy"

_cancers, 2022, doi:10.3390/cancers14071830_

Round 1
Reviewer 1 Report
Overall the work is excellent, it's been a pleasure to read. Biologically on the point. Congratulations to the authors.
I have but only minor comments:
line 77 & 125 : I suggest to replace "cancer mutation" with "cancer-associated mutations"
line 107 & 120 : Signaling scaffolds increase local concentration" "higher level of signalling"
Some detail about what concentrations in pM/uM, and what imaging technique was used to detect signaling is desirable.
Here or somewhere else in the manuscript, some discussion about what technique can detect those small concentrations would be nice.
line 142 : "the" transcription of oncogenic... ?
Around line 286-290 : As the authors mentions, the distribution of drugs of cells is highly heterogeneous and other give example of tamoxifen in breast cancer. Hence, a discussion about the methods and imaging technique that are available to monitor the drug activity and distribution of active compounds or metabolized compounds in cells would be of interest for readers. I expect it would help people to design studies on this particular aspect. Would be localized mass-spectrometry of use ? How about label-free spectroscopy techniques ? A few studies have shown for example the use of Raman spectroscopy combined with mass-spectrometry to monitor tamoxifen action in HepG2 cells, it seems relevant in this case to discuss such approaches to provide hints of possible approaches.
Author Response
R1
Overall the work is excellent, it's been a pleasure to read. Biologically on the point. Congratulations to the authors.
Thank you very much for this encouraging comment.
I have but only minor comments:
line 77 & 125 : I suggest to replace "cancer mutation" with "cancer-associated mutations"
We have done as suggested
line 107 & 120 : Signaling scaffolds increase local concentration" "higher level of signalling"
Some detail about what concentrations in pM/uM, and what imaging technique was used to detect signaling is desirable.
Here or somewhere else in the manuscript, some discussion about what technique can detect those small concentrations would be nice.
We apologize for using the term concentration when referring to signaling scaffolds. Scaffolds represent single molecular entities and it is not appropriate to use the term concentration that refers to number of molecules/volume. We have corrected that in the text. It is also worth further discussing the role of protein concentration in LLPS.
The thermodynamics of phase separation for polymers is described by the Flory-Huggins Model that describes the free energy for mixing a polymer and a solvent and therefore specifies the conditions for demixing or phase separation. The protein concentration found in demixed biological condensates has been modeled in vitro and found to depend on both the temperature and salt concentration (Brady et al. 2017). Demixing is favored by lower temperatures and lower salt concentrations. Phase separation is then represented as a phase diagram curve that shows the proportion of demixed polymer as a function of their concentration, temperature and salt concentration. Hence, demixing occurs within a range of concentrations. For the RNA binding protein Ddx4 condensates were observed at around 5 mM Ddx4 in 200 mM NaCl. However, mutants of Ddx4 in their RG and FG motifs do not phase separate in these conditions (Brady et al. 2017). This study illustrates the importance of sequence properties mediating intermolecular interactions for demixing. The protein concentration in the phase separated state can be estimated using 1H-15N heteronuclear quantum coherence spectrum. For the RNA binding protein FUS the estimated value was 27.8 mM (Murthy et al. 2019). A similar concentration (around 14 mM) was obtained using Raman spectroscopy of condensates formed by ataxin-3, a protein that forms aggregates in the neurodegenerative Machado–Joseph disease (Murakami et al. 2021). We added the following text:
The protein concentration in phase separated structures depends on the ability of each protein to engage in intermolecular interactions. For example, the RNA binding protein DDX4 forms condensates at a concentration close to 5 mM but a mutant of this proteins that cannot form interactions does not demix in the same conditions (Brady et al. 2017). A direct measurement of the concentration in the condensates using 1H-15N heteronuclear quantum coherence spectrum showed a much higher value of 27.7 mM for the RNA binding protein FUS (Murthy et al. 2019). A similar concentration (around 14 mM) was obtained using Raman spectroscopy of condensates formed by ataxin-3, a protein that forms aggregates in the neurodegenerative Machado–Joseph disease (Murakami et al. 2021). The total protein concentration in cells has been estimated around 2.5 mM (Ellis 2001). Therefore the degree of crowding in biomolecular condensates is not too different from the rest of the cells but the effective increase in concentration for each phase separated protein is considerable.
rline 142 : "the" transcription of oncogenic... ?
We have done as suggested
Around line 286-290 : As the authors mentions, the distribution of drugs of cells is highly heterogeneous and other give example of tamoxifen in breast cancer. Hence, a discussion about the methods and imaging technique that are available to monitor the drug activity and distribution of active compounds or metabolized compounds in cells would be of interest for readers. I expect it would help people to design studies on this particular aspect. Would be localized mass-spectrometry of use ? How about label-free spectroscopy techniques ? A few studies have shown for example the use of Raman spectroscopy combined with mass-spectrometry to monitor tamoxifen action in HepG2 cells, it seems relevant in this case to discuss such approaches to provide hints of possible approaches.
The pioneering paper reporting localized distribution of cisplatin in cells used multiplexed ion beam imaging (MIBI) and a Nano SIMS (Nanoscale secondary ion mass spectrometry) instrument. We have now indicated that this was the technique used. We also mentioned the suggested work using Raman spectroscopy to study the effects of tamoxifen at a single cell level.
Reviewer 2 Report
The review article “Liquid-liquid phase separation in cancer signaling, metabolism and anticancer therapy” authored by Sebastian Igelmann, Frédéric Lessard, Gerardo Ferbeyre summarizes recent progress in the study on how liquid-liquid phase separation plays roles in cancer development and anti-cancer therapy. The manuscript covers a wide range of topics published in recent papers. As liquid-liquid phase separation has been gaining a large interest in many biological processes, this review article is highly timely and provides a lot of useful information to researchers not only in cancer biology, but also in general cell biology. I would like to recommend this for the publication in Cancers. I have only a few minor comments.
- A paragraph in section 1.4 (line 225-233) is more or less irrelevant to the main topic of the manuscript. I agree that it is important to mention about pivotal roles of liquid-liquid phase separation in bacteria and plants, it may not be directly linked to cancer. The authors should re-consider this paragraph.
- It may be more informative to some expertized readers if the authors provide more detail information about the “condensates” described in individual papers. It had been demonstrated that “condensate” has a large variety of properties, including mobility, reversibility etc. In the case of neuro-degenerative diseases, for example, the mobility of the “condensate” has been changed by amino acid mutations, although both cases can be classified as “condensate”. Therefore it may be useful to describe liquid properties of condensates or membrane-less organelle that they describe in the manuscript (not all of them but at least some of them if it is described in the original articles).
Author Response
The review article “Liquid-liquid phase separation in cancer signaling, metabolism and anticancer therapy” authored by Sebastian Igelmann, Frédéric Lessard, Gerardo Ferbeyre summarizes recent progress in the study on how liquid-liquid phase separation plays roles in cancer development and anti-cancer therapy. The manuscript covers a wide range of topics published in recent papers. As liquid-liquid phase separation has been gaining a large interest in many biological processes, this review article is highly timely and provides a lot of useful information to researchers not only in cancer biology, but also in general cell biology. I would like to recommend this for the publication in Cancers. I have only a few minor comments.
1. A paragraph in section 1.4 (line 225-233) is more or less irrelevant to the main topic of the manuscript. I agree that it is important to mention about pivotal roles of liquid-liquid phase separation in bacteria and plants, it may not be directly linked to cancer. The authors should re-consider this paragraph.
We agree is not linked to cancer but we included those examples to show how accessory proteins determine phase separation for these metabolic complexes. Hence, the need for a complete proteomic characterization of any cancer associated metabolome because the key components for phase separation may not be one of the enzymes in the pathway.
2. It may be more informative to some expertized readers if the authors provide more detail information about the “condensates” described in individual papers. It had been demonstrated that “condensate” has a large variety of properties, including mobility, reversibility etc. In the case of neuro-degenerative diseases, for example, the mobility of the “condensate” has been changed by amino acid mutations, although both cases can be classified as “condensate”. Therefore it may be useful to describe liquid properties of condensates or membrane-less organelle that they describe in the manuscript (not all of them but at least some of them if it is described in the original articles).
Thanks for this suggestion. As noticed by the reviewer, our initial description of oncogenic condensates did not precisely describe the mobility of most of them. We described the solid like state (limited mobility) of the EML4-ALK condensates but not the others. As a consequence it was not clear what is the general state of all newly described condensates in cancer. We now added FRAP data for SHP2, NF2 and ENL condensates. The data show a liquid behavior which is more consistent with a signal modulation role for biomolecular condensates as we propose rather that sequestration models suggested by others. We could not find FRAP data for all condensates, for example NUP98 or EWS-FLI1 condensates were not analyzed by FRAP.